# Impact of Moisture Content on the Elasto-Viscoplastic Behaviour of Rammed Earth Wall: New Findings

**Taini Chitimbo [1],\*, Feras Abdulsamad [1,2,3], Noémie Prime [1],\*, André Revil [1,2,3] and Olivier Plé [1]**

1   UMR CNRS 5271, LOCIE, Université Savoie Mont-Blanc, 73370 Le Bourget du Lac, France
2   UMR CNRS 5204, EDYTEM Laboratory, Université Grenoble Alpes, 73370 Le Bourget du Lac, France
3   UMR CNRS 5204, EDYTEM Laboratory, Université Savoie Mont-Blanc, 73370 Le Bourget du Lac, France
\*   Correspondence: taini.chitimbo@univ-smb.fr (T.C.); noemie.prime@univ-smb.fr (N.P.)

**Abstract:** The influence of hydric state on the elasto-viscoplastic behaviour of a unstabilised rammed earth (URE) wall has yet to be studied in the literature. This paper presents an experimental campaign on a rammed earth wall. The aim is to evaluate the link between the mechanical properties (including viscosity) and the varying hydric state inside the drying wall after manufacture. Cyclic axial compression and stress relaxation tests were carried out for this purpose. A compression test was conducted up to 0.1 MPa, followed by a stress relaxation test. These tests were periodically performed over 32 weeks. In addition, the hydric state inside the wall was monitored by humidity sensors. The results show that both the elastic modulus and the dynamic viscosity coefficient increase as the structure dries. A dependence of the mechanical behaviour on time is therefore found in these samples in the transient state. This can occur when the sample is in the drying or wetting phase. As rammed earth is a material particularly sensitive to water, this result is crucial for the durability of earthen constructions.

**Keywords:** unstabilised rammed earth (URE); compressive strength; viscosity; relative humidity (RH)

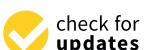



## 1. Introduction

Due to the rapid growth of interest in earth as a construction material, research on these materials has been increasingly important. There are several earth construction techniques, among which rammed earth (RE) has been used since historical times. When no chemical binder is added to RE, it is known as unstabilised rammed earth (URE). URE is known to be a sustainable option for material construction due to its low embodied energy, its subsequent low Green House Gases emission [1], interesting hygro-thermal properties, low operational cost, recyclability [2] and, historically, its relatively good durability [3]. Finally, this material appears to have a compressive strength of between 0.5 and 4 MPa [4], which is sufficient for the structure of a building with a few storeys.

URE is a compacted soil, its mechanical strength can be described by its compressive strength and its Young's modulus. Compressive strength is a widely used mechanical parameter that dictates the choice of material for building construction [5]. There are different standards, suggesting the minimum characteristics of compressive strength ranging between 0.4–2 MPa [6–8]. These values are similar to those found in the literature, ranging between 0.5–4 MPa [9–14]. The significant variation is due to: the water content during the test, compaction energy, dry density, particle size distribution, clay content, and sample shape.

For the Young's modulus, the recommended design value ranges between 150–500 MPa. However, the values of the Young's modulus reported in the literature have a high dispersion, which is related to several factors: the heterogeneity of the specimens and the method of measurement.

In addition, this work will add a specific focus on the viscous part of the behaviour. It is known that the mechanical behaviour of URE can vary with their hydric state, and,

therefore, with climatic conditions [15], notably during the initial drying period after wet compacting [16]. Various studies have highlighted the sensitive nature of URE to moisture conditions [11,12,17–19].

Beckett et al. [17] and Xu et al. [19] conditioned URE samples under varying relative humidity and temperatures to investigate the mechanical response at each condition.

In both studies, samples were conditioned at different temperatures between 15 °C and 40 °C and different RH (23% and 97%). Becket et al. [17] reported that the compressive strength at low RH (30%) was almost twice that at high RH (90%). Xu et al. [19] found that both the maximum deviatoric stress and the Young's modulus decrease with the increase in relative humidity. On the other hand, in both studies, there was only a slight influence of temperature for a fixed RH. Therefore, it was concluded that RH had a greater influence on mechanical performance than temperature.

When samples are conditioned, the moisture distribution is uniform and, therefore, does not represent the reality, in which the earth never reaches, throughout its mass, a total equilibrium with the ambient conditions. Other studies dried the samples at atmospheric conditions to investigate the impact of moisture on mechanical behavior [16,20,21].

Bui et al. [20] conducted an unconfined compressive strength test on RE samples at a wide range of moisture contents. Samples were dried at atmospheric conditions with moisture contents varying from manufacturing (11%) to dry states (1–2%). The results show that the strength and the Young's modulus increased by a factor of four from the manufacturing state to the dry state. Similarly, Chitimbo et al. [16], after drying samples at ambient conditions, observed an increase in the mechanical performances by a factor of ten, from the moist state to the dry state.

From these studies, it is clear that the compressive strength and Young modulus increases with the decrease in water content.

Despite the vast span of research devoted to this subject, most of the existing studies tend to be more focused on the elastic mechanical properties (Young's modulus) and compressive strength; to our knowledge, there are no existing studies on the visco-plastic behaviour of URE, which depends on time and hydric state. Gil-Martín et al. [22] seem to be the first to show a degradation in the elastic stiffness of URE when studying the creep effect for 15 days.

The viscous behaviour and its dependency on the hydric state have a significant influence on the performance of URE buildings. Therefore, a good understanding of this behaviour is of vital importance for the practical utilisation of URE. In addition, only a few studies have been conducted on a large scale of earth structure (see Table 1). There is a lack of results at a large scale, although the mechanical capacity at this scale could be slightly different due to scale effect. For example, Bui et al. [23] performed tests at three different scales (compressed earth blocs, structural columns element, and wall scale). The results of the modulus of elasticity calculated from the small-scale samples were close to each other, while for the wall scale, higher results than the other two approaches were reported. Thus, it's clear that there is a significant lack of experimental results at the wall scale, which is closer to the real building scale.

**Table 1.** Summary of experimental studies of mechanical behaviours of URE at large scale.

| Specimen Size | $\rho$d (kg/m$^3$) | Rc (MPa) | *E* (Mpa) | References |
|---|---|---|---|---|
| $1.0 \times 1.0 \times 0.3$ m$^3$ | Not presented | 0.6–0.7 | 60 | [11] |
| $1.0 \times 1.0 \times 0.3$ m$^3$ | 2000 | $1.3 \pm 0.2$ | $500 \pm 40$ | [24] |
| $0.55 \times 0.55 \times 0.2$ m$^3$ | 2100 | 1.26 | 1034 | [25] |
| $2.5 \times 2.5 \times 0.5$ m$^3$ | 1920 | 0.37 | – | [26] |
| $1.5 \times 1.5 \times 0.25$ m$^3$ | 1780–1850 | | 400 | [14] |

This paper presents experimental results concerning the mechanical behaviour of URE at a wall scale (1/2 scale of a real wall), with height H = 1.5 m, width W = 0.75 m and thickness b = 0.2 m. The viscous properties of the URE wall and its evolution with drying time, and therefore with global hydric state, are presented. Finally, the scale effect on the mechanical behaviour of URE is studied.

*Background on Rheology of Soil Materials*

Elasto-viscoplasticity is a behaviour that describes strain rate-dependent non-recoverable behaviour. Creep and relaxation phenomena are consequences of this elasto viscoplastic behaviour of soil. Stress relaxation is the reduction in stress with time, under constant strain. These phenomena are due to a re-arrangement of the material on the molecular- or micro-scale [27].

Perzyna [28] proposed some preliminary models, which were based on elasto-viscoplastic framework, to study this behaviour of soil. Furthermore, the visco-elastic behaviour can be modelled by using basic elements such as spring and dashpot arranged in series, known as the Maxwell 1D model (Figure 1a), or in parallel, also known as the Kelvin Voight model (Figure 1b). Differential equations for these models are given by Equation (1). Where $\sigma$ is stress, $\varepsilon$ is strain, $E$ is Young's modulus and $\eta$ is the coefficient of viscosity.

$$\dot{\varepsilon}(t) = \frac{\dot{\sigma}(t)}{E} + \frac{\sigma(t)}{\eta} \qquad \sigma(t) = E\varepsilon + \eta\dot{\varepsilon}(t) \tag{1}$$

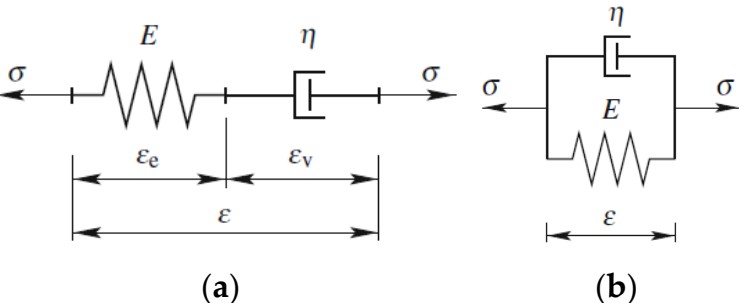

**Figure 1.** (**a**) Maxwell model and (**b**) Kelvin Voight model with their respective stress-strain rate Equations.

Although the Maxwell model can describe stress relaxation, after a sufficiently long time period, the stress will relax to zero, which is not realistic for a humid URE structure. In order to describe more realistically the stress relaxation as well as the creep, more complicated models, combining several springs and/or dashpots, are required. Among these, the three-element model of Figure 2a is the simplest, allowing the stress to relax to a non-zero value (Figure 2a). According to this model, the Young's moduli $E_1$ and $E_2$ remain constant over time. It means that, $E_1$ is the initial Young's modulus of material before relaxation and $E_2$ is attained when the relaxation reaches a constant value. The differential constitutive relation for this model is given by Equation (2) [27].

$$\sigma + \frac{\eta}{E_1 + E_2}\dot{\sigma} = \frac{E_1 E_2}{E_1 + E_2}\varepsilon + \frac{\eta E_1}{E_1 + E_2}\dot{\varepsilon} \tag{2}$$

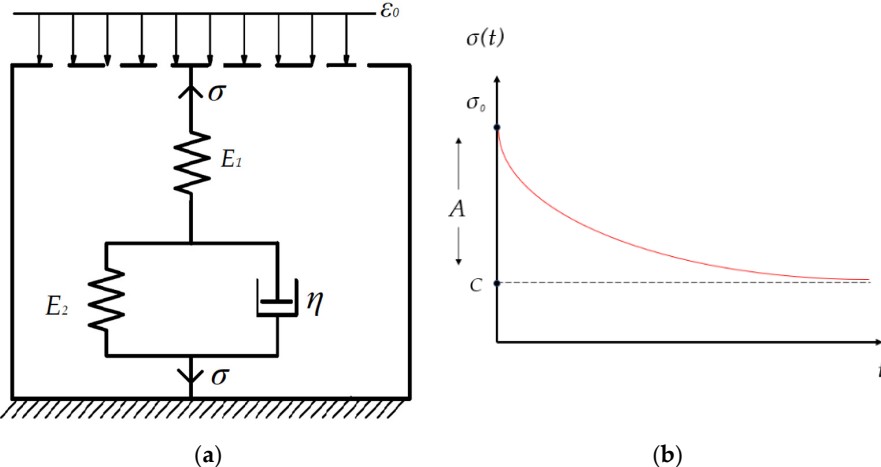

**Figure 2.** (**a**) Three-element model to describe the elasto-viscoplastic behaviour. (**b**) Stress relaxation function.

In the relaxation test, the wall is subjected to a constant strain $\varepsilon_0$ at $t$ = 0. Then, $\dot{\varepsilon} = 0$; therefore Equation (2) becomes:

$$\sigma + \frac{\eta}{E_1 + E_2}\dot{\sigma} = \frac{E_1 E_2}{E_1 + E_2}\varepsilon \tag{3}$$

$$\sigma + \frac{\eta}{E_1 + E_2}\dot{\sigma} = Constant \tag{4}$$

$$\dot{\sigma} = -\frac{E_1 + E_2}{\eta}\sigma + Constant \tag{5}$$

The solution of the differential Equation (5) is given by:

$$\sigma(t) = Ae^{-(\frac{E_1+E_2}{\eta})t} + C \tag{6}$$

Equation (6) is the stress relaxation function and its curve, illustrated in Figure 2b. Where $A$ and $C$ are constants. By substituting Equation (6) into Equation (2), we obtain $C = \frac{E_1 E_2}{E_1 + E_2}\varepsilon$. At time $t$ = 0, Equation (6) becomes $\sigma_0 = A + C$; hence, constant $A$ is given by $A = \sigma_0 - C \cdot \eta$.

## 2. Materials

### 2.1. Soil Properties

The earth studied was excavated from Saint Quentin Fallavier in Auvergne-Rhone-Alpes region, in Southeastern France, and in 2019, was used to build the Orangery Building in Confluence, Lyon. The soil contains approximately 10% clay, 12% silt, 33% sand and 45% gravel. The clay part was further classified as slightly active clay; hence, the soil is less sensitive to shrinkage. More on the earth mineralogical composition can be found in Chitimbo et al. [16]. To describe the affinity with water, the earth's water retention properties are essential. The soil water retention curve (SWRC) represents a fundamental constitutive relationship in unsaturated soil mechanics. Specifically, it describes the relationship between the soil suction and water content (gravimetric or volumetric) or the degree of saturation of the porous media. Here, two methods have been implemented for this: the filter paper method and the saline solution method at 25 °C. Sorption and desorption paths have been performed. The earth's water retention properties were measured by Chitimbo et al. [16] and then fitted by using the Van Genuchten relationship, which is described as:

$$\theta(s) = \theta r + \frac{\theta s - \theta r}{\left[1 + (\alpha|s|)^n\right]^m} \tag{7}$$

where $\theta$ is the volumetric water content, $s$ is suction pressure (MPa), $\theta s$ is saturated water content, $\theta r$ is residual water content, $\alpha$ (MPa)$^{-1}$, $n$ and $m$ are fitting parameters. The fitting parameters were evaluated as $\alpha$ = 0.39 MPa$^{-1}$, $n$ = 1.7, and $m$ = 0.411. The results are presented on the graph in Figure 3.

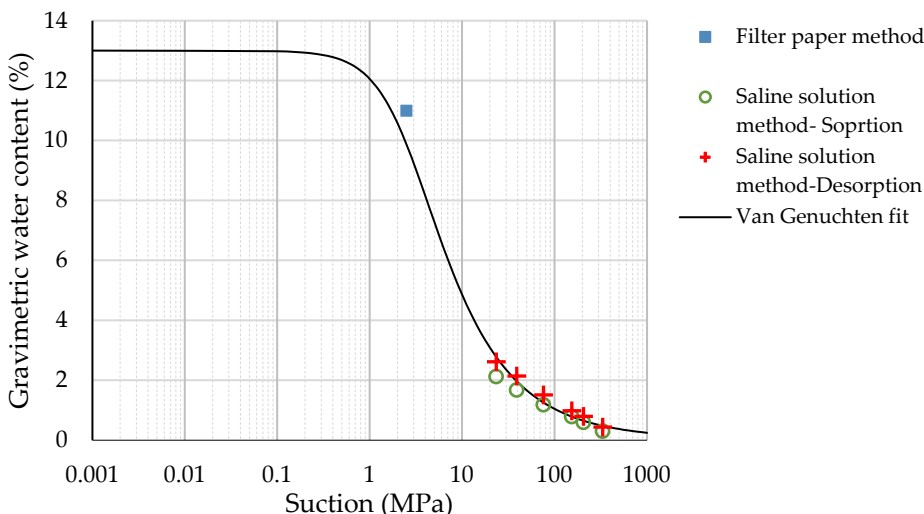

**Figure 3.** Soil water retention curve (SWRC) using saline solution method, filter paper method, Van Genuchten models at 25 °C [16].

*2.2. Wall Manufacturing*

Two rammed earth walls of size 1.5 × 0.75 × 0.2 m$^3$ were manufactured at the LOCIE laboratory. They will be named 'wall 1' and 'wall 2' in what follows. Compaction was conducted in a metal formwork and plywood on the sides (Figure 4a). Raw earth was mixed with water and stored in a covered container for one day before the fabrication to assure the homogenisation of the water content. Moist soil was obtained with an optimum gravimetric water content of 10%, with respect to the normal proctor test. It was poured into the formwork and dynamically compacted (using a hydraulic hammer) in 20 layers (Figure 5). Each compacted layer had a thickness of approximately 7 to 8 cm, according to the guidelines and standards. In France, the practical guide for construction using rammed earth is Guide de bonnes pratiques (GBP) Pisé [29]. There are no regulations on the number of layers for samples. After compaction, the average dry density was 1865 kg·m$^{-3}$. Finally, the top layer was levelled to obtain a smooth surface, the formwork was removed (Figure 4b) and a wood beam was placed on it (Figure 4c).

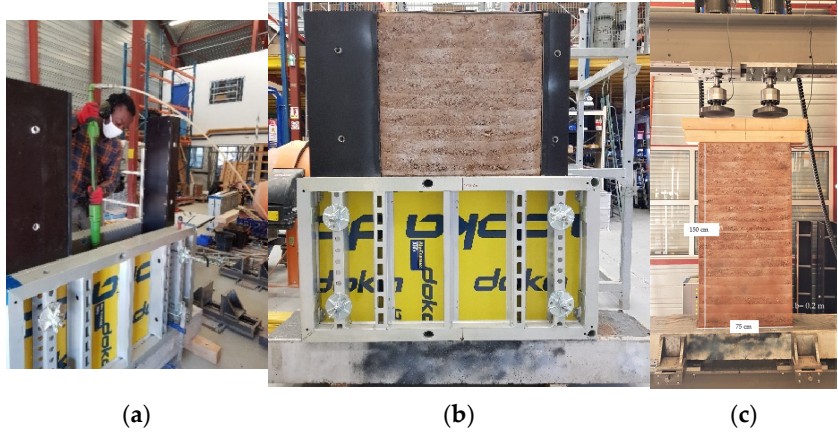

|     (a)     |     (b)     |     (c)     |

**Figure 4.** (**a**) Compaction of earth in a formwork using a pneumatic hammer. (**b**) Removing formwork after compaction, compacted earth layer can be observed. (**c**) URE wall subjected to mechanical test using two electric actuators.

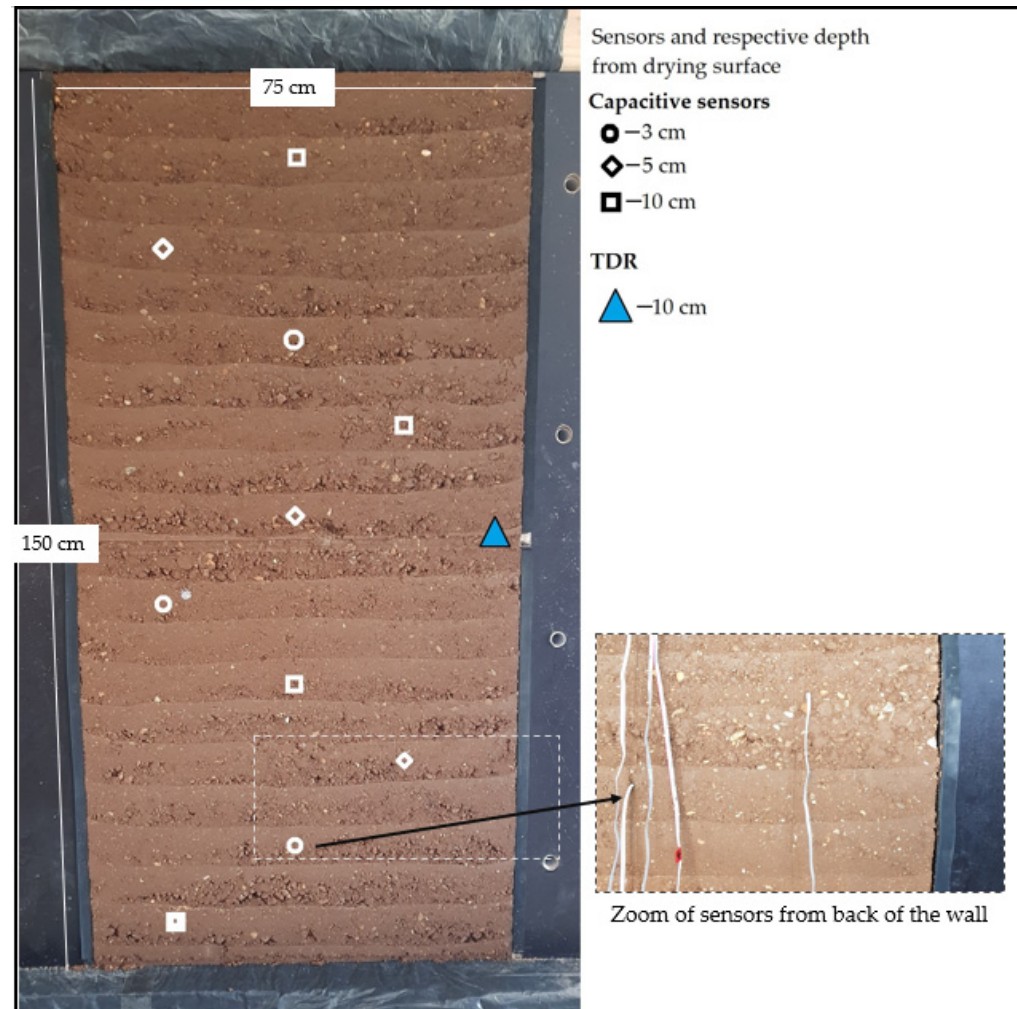

**Figure 5.** Layout of position of the sensors with their respective depth in the wall. A zoom of sensors final position taken from the back of the wall.

## 3. Experimental Protocol

### 3.1. Hydric Monitoring

The kinetic of drying for wall 2 was followed by relative humidity sensors (capacitive sensors) and a water content sensor (Time Domaine Reflectometer, TDR). Figure 6b shows the position of these sensors on the wall. The capacitive sensors used were SHT75, covered by a porous polyethylene material to protect it from damage (Figure 6a).

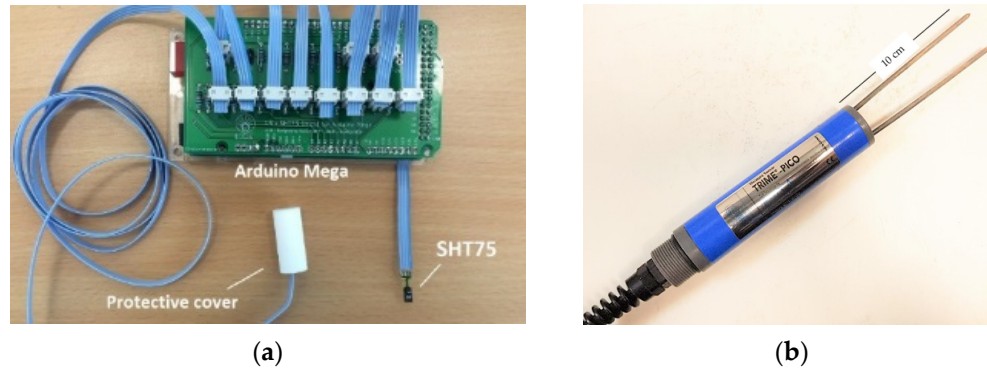

(**a**)          (**b**)

**Figure 6.** (**a**) Capacitive sensors, (**b**) Time Domaine Reflectometer.

The capacitive sensors were first calibrated by using a cooled mirror hygrometer in a calibration chamber for both relative humidity and temperature, with ranges of 5 °C/30 °C for temperature and 9%/98% for relative humidity. The calibrated sensors were placed in the sample during manufacturing. They were covered by a polyethylene filter as protection from damage. In each sample, a total of nine sensors were used. These sensors were placed in the middle of each layer, at a distance of 3 ± 0.5 cm, 5 ± 0.5 cm and 10 ± 0.5 cm from the drying surfaces. The TDR probe used in this study was TRIME-PICO-32 (see Figure 6b), which provides a measurement of the volumetric water content with an accuracy of ±2%. The device was calibrated by measurements on the saturated sand and compacted soil of the known volumetric water content. After calibration, the TDR was placed until a depth of 10 cm, on the right side at the centre of the wall.

### 3.2. Mechanical Testing (Elasto-Viscoplastic Behaviour)

In order to determine the compression strength, wall 1 was tested directly after being manufactured, in its moist state (w = 10%). Wall 2 was dried in the laboratory hall for 32 weeks. Mechanical tests were performed to determine the elastic and viscous behaviour, as described in Figure 7. Wall 1 was tested before failure, in the moist state, whereas wall 2 was not tested at time 0 (in order to avoid damage at such a weak state) but from one week up to 32 weeks. The displacement controlled mechanical test is adopted. The loading path is illustrated in Figure 7: a first load until 0.1 MPa was performed (OA in Figure 7), then an unloading-reloading up to 0.1 MPa was conducted to determine the elastic behaviour (ABC); lastly, the displacement is kept fixed for 100 to 500 s in order to determine the viscous component of the behaviour in relaxation (CD). The maximum of loading, 0.1 MPa, both prevents damage on the wall and is representative of a classical vertical apparent stress for a one-story building. The loading was controlled with a constant displacement of 1.2 mm/min. In Figure 7, points OAB represent the loading-unloading path with corresponding plastic and elastic deformation. The strain was calculated using field displacement through the Digital Image Correlation (DIC) technique in the central part of the wallet in order to avoid border effects between state A and B (see Figure 7).

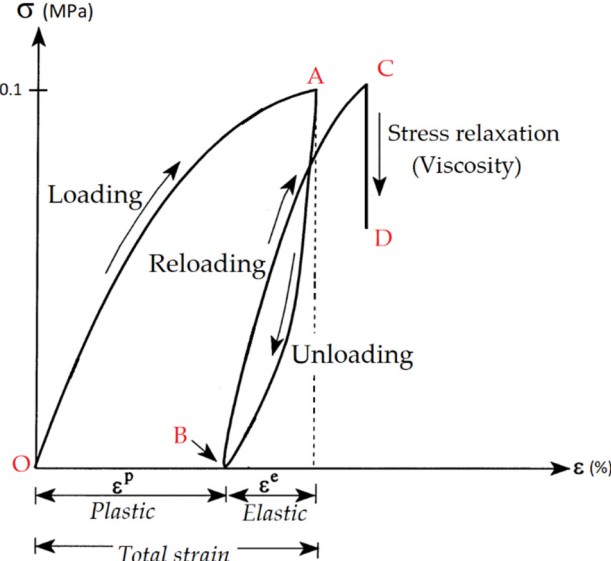

**Figure 7.** Loading-Unloading cycle represented by path OAB, and stress relaxation represented by path CD.

Viscous behaviour was determined from a stress relaxation test. This test shows the dependency of stress with time, while the displacement is fixed. The relaxation test is represented by points C and D (see Figure 7) and its corresponding stress-time dependency.

A rheological model, represented in Figure 2, was adopted in order to predict the viscous behaviour of the URE wall; this behaviour was modelled using Equation (6).

## 4. Results

### 4.1. Hydric State

With the help of the capacitive sensors, the relative humidity was tracked in wall 2. Figure 8a presents the evolution of RH with time for the wall dried for eight weeks. It was observed that no variation of RH can be measured from the first three weeks, although it is where the mass loss seems to be more intense, according to the TDR results (see Figure 8b). This was due to the limitation of the capacitive sensors: in very humid conditions (RH > 98%), they can hardly detect changes in relative humidity [30]. After three weeks of drying, a constant decrease in RH was recorded from each sensor. It was also observed that the sensors placed near the drying surfaces recorded a higher drying rate and responded quickly to the change in environmental humidity. Figure 8b shows the evolution of the water content determined by TDR. The water content decreased from a manufacturing water content of approximately 10% to 4.5% in eight weeks.

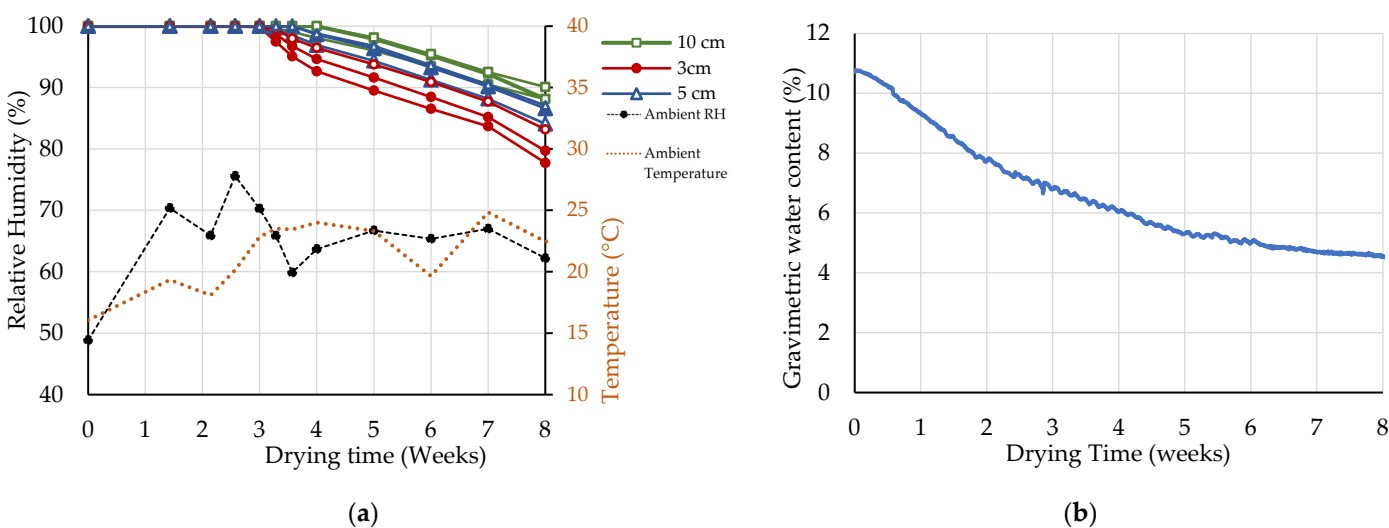

| (**a**) | (**b**) |

**Figure 8.** (**a**) Evolution of RH by capacitive sensors, (**b**) Evolution of water content measured by TDR.

### 4.2. Compressive Strength and Apparent Young's Modulus

The mechanical resistance was determined on wall 1 in the moist state and on wall 2 after 32 weeks of compression, until rupture. At the initial moist state, the compressive strength found was 0.1 MPa, and at a dried state it was found to be 1.15 MPa. The apparent Young's modulus was determined from the unloading path AB (see Figure 7); the results are summarised in Figure 9. The results show that the apparent Young's modulus of the walls increased from 190 MPa to 1020 MPa after eight weeks of drying from manufacture and then to 1024 MPa after 32 weeks. It was observed that the results of the evolution of the apparent Young's modulus are very comparable to those obtained on the URE columns (see Figure 9) by Chitimbo et al. [16], for very comparable materials (soil composition hence same granulometry) and densities (1865 $\pm$ 39 kg·m$^{-3}$ for the wall and 1893 $\pm$ 39 kg·m$^{-3}$ for columns). According to the Guide de bonnes pratiques (GBP) Pisé, 2018, a URE column can be considered as a representative elementary volume.

### 4.3. Viscous Behaviour

Figure 10 shows the evolution with drying weeks of the stress relaxation curve during phase CD of the loading path (wall 1 at $t$ = 0, and wall 2 for one weeks and after). It was observed that, on each curve, the stress reduces exponentially. For eight weeks, the relaxation has been realised for a sufficient time to put into evidence that stress tends to an asymptotic value. Using the three-element model for relaxation from Equation (6), the

behaviour was modelled for each week and represented by the dashed line in Figure 10. From the model, the dynamic viscosity was determined (see Figure 11). An increase in the coefficient of viscosity ($\eta$), from $10^9$ to $10^{11}$ Pa·s, during eight weeks of drying, is found. A significant change of $\eta$ by a factor of 100 suggests the viscous behaviour should be considered in modelling the coupled hydro-mechanical behaviour of URE. When the wall is dried, the value of the viscosity ranged between, approximately, $1.41 \times 10^{11}$ Pa·s. A similar order of magnitude in the coefficient of viscosity ($4.834 \times 10^{12}$ Pa·s) was reported by Gil-Martín et al. (2022), who studied a RE cylindrical column ($\Phi$ = 15 cm and H = 30 cm) during a creep test using the same rheological model (see Figure 2a). From Figure 11, it is clear that the viscous behaviour of rammed earth is highly dependent on the hydric state. In the real building, viscosity can take place when the wet earth is loaded, and strain can develop by creep.

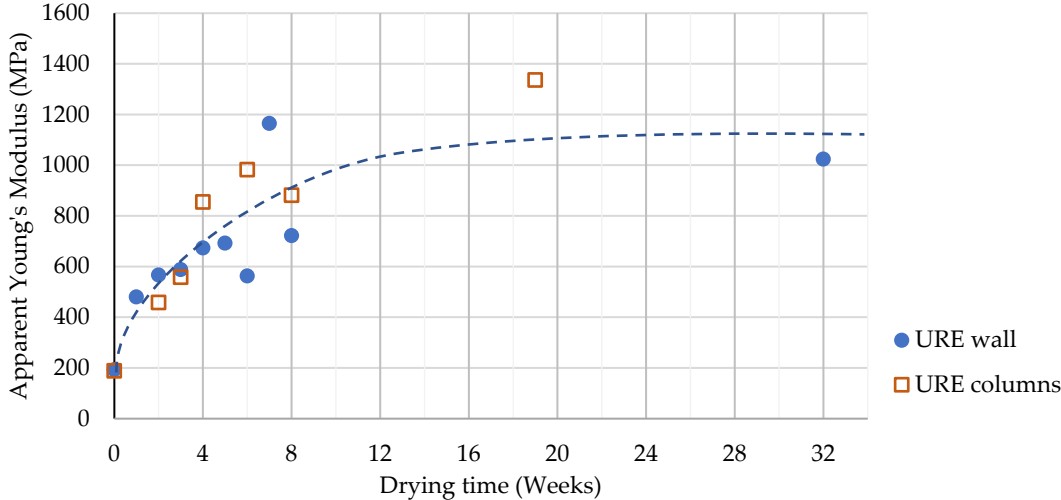

**Figure 9.** Evolution of Apparent Youngs's modulus of URE wall with drying time along with its tendency shown by dashed line. The results of wall are compared to the one of columns prepared by using similar soil and have similar dry density [16].

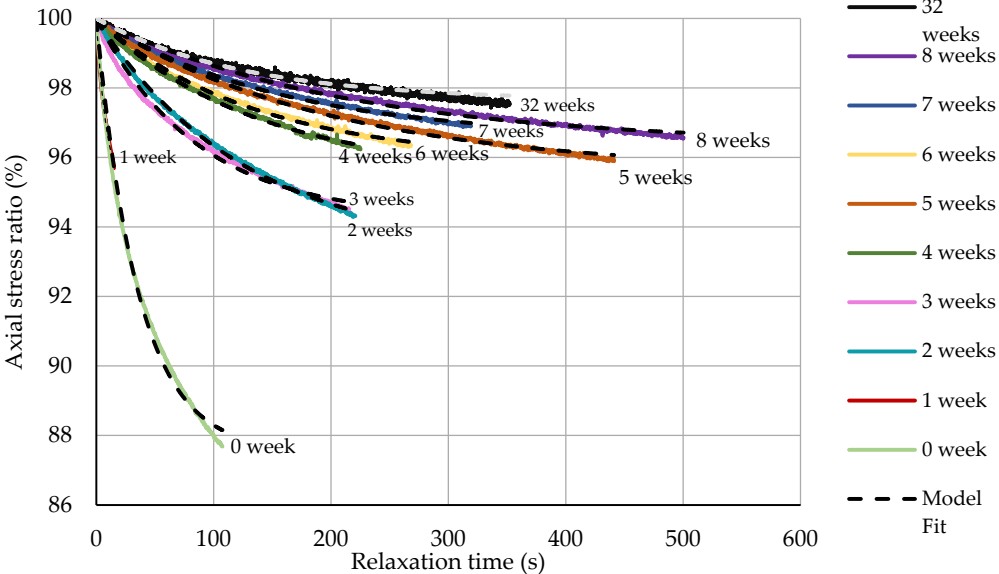

**Figure 10.** Stress relaxation test results showing experimental and modelled curve at each week on a drying URE wall.

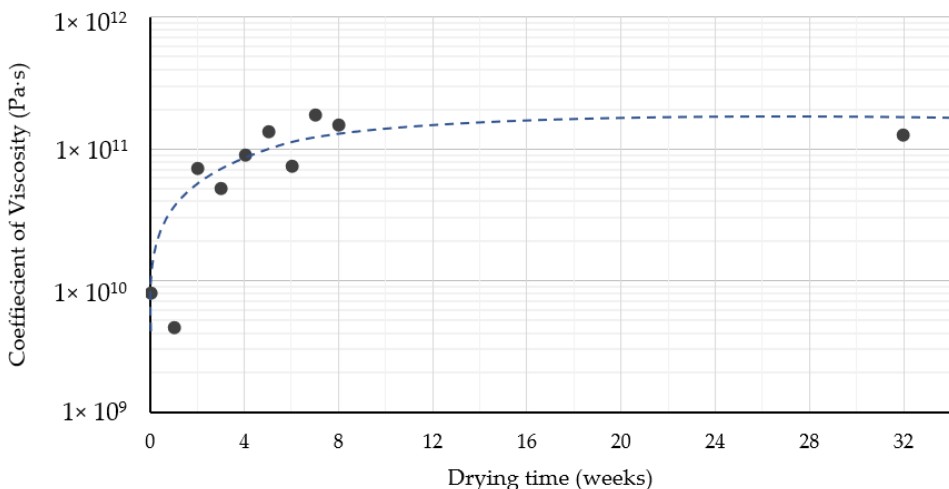

**Figure 11.** Evolution of coefficient of dynamic viscosity with drying time, along with its tendency shown by dashed line.

## 5. Discussion

### 5.1. Study of Suction as a State Variable to Represent Hydric State

Suction has been shown to be a very suitable parameter in order to understand and model the unsaturated behaviour of the earth's porous medium [21]. It is thus important to adopt it as a single variable to stand for the hydric state.

The RH (Figure 8) and their corresponding temperatures were expressed in suction by using the Kelvin's Equation (8) [31]. The Kelvin's equation is defined as:

$$s = -\frac{\rho_w \cdot R \cdot T}{M_w} \ln(\text{RH}) \tag{8}$$

where $s$ is the suction at a given temperature $T$ (in Kelvin, K), $R$ is the universal gas constant ($R = 8.3143$ J·mol$^{-1}$·K$^{-1}$), $M_w$ is the molar mass of water ($M_w = 0.018$ Kg/mol), $\rho_w$ is the bulk density of water ($\rho_w = 1000$ Kg/m$^3$) and RH is the relative humidity which is defined as the ratio of partial vapor pressure P in the considered atmosphere and the saturation vapor pressure Po at a particular temperature ($T = 298$ K).

The suction values were computed locally at depths of 3 cm, 5 cm, and 10 cm (position of capacity sensors) (Figure 12). In addition, suction values deduced from the TDR, as shown in Figure 11, were obtained from the evolution of the water content, thanks to the Van Genuchten model of water retention curve (Figure 3).

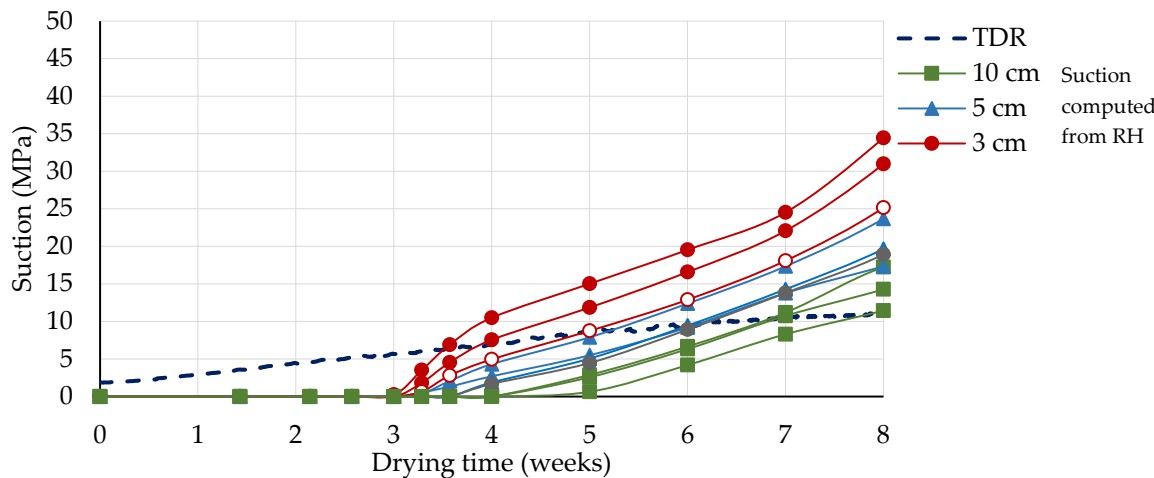

**Figure 12.** Evolution of suction in RE wall deduced from RH and water content evolution.

These curves show a consistent evolution, except for the first 3 weeks on which the SHT sensors cannot measure the very slight evolution of RH, responsible for a not negligible variation of water content once Kelvin equation is used.

### 5.2. Total Mechanical Properties

The evolution of the total mechanical behaviour was linked to the hydric state within the wall, as presented by its suction values. Figure 13a shows the results of the apparent Young's modulus with the corresponding suction. In this work, it was observed that the logarithm of suction is linearly correlated to the apparent young's modulus; a similar relation was reported by Bui et al. [20] and Chitimbo et al. [16]. The results of this work at a wall scale are compared with the one at a columns scale using similar soil and drying methods. Figure 13a shows the results of these two scales. A difference was observed in the evolution of the apparent Young's modulus with respect to the suction. However, this difference is due to large uncertainty in the suction (Figure 12), as the specimens didnot dry uniformly. On the other hand, the coefficient of viscosity was also linked to the suction (Figure 13b); a linear relation, with a good correlation for eight weeks, was shown, and similar results are observed for the soil material containing clay [32].

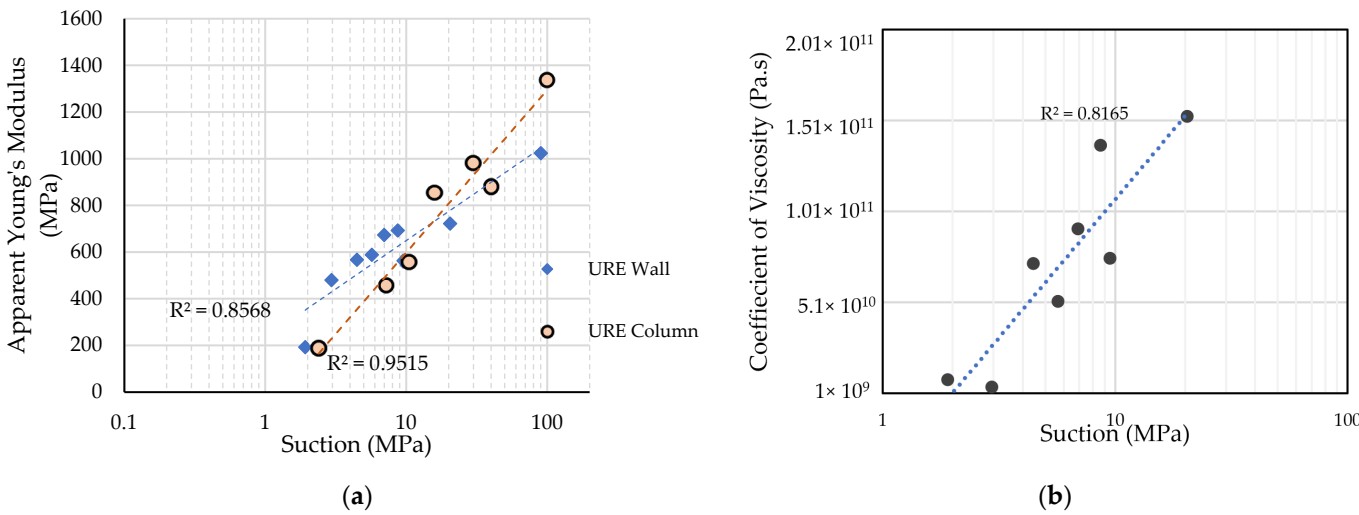

**Figure 13.** (**a**) Evolution of apparent Young's modulus with suction at wall scale (this study) and column [16], (**b**) Evolution of Coefficient of viscosity with the logarithm of suction for URE wall.

### 6. Conclusions

In this paper, the mechanical behaviour of URE has been studied at a wall scale close to a real building scale. The characterisation of the earth used is presented, containing 10% of clay, which is lowly sensitive to shrinkage. The URE walls were dried after being manufactured and, while drying, periodic monitoring of the hydric state was performed. Thus, the evolution of the water content, as well as the RH, were obtained.

The walls were then subjected to mechanical tests, including Loading-unloading cycles (elastoplastic behaviour) and stress relaxation (viscous behaviour). These tests were performed each week for eight weeks of drying and then at the dried state; thus, the evolution of the mechanical behaviours with respect to time and hydric state were presented. Major contributions from this paper include highlighting the viscous behaviour of URE and its evolution with the hydric state. Considering that none of the existing studies have ever shown the variation in the visco-elastoplastic behaviour of URE, this paper will pave the way for taking into consideration the viscous part of the mechanical behaviour of URE. In particular, it provides the order of magnitude for the coefficient of viscosity ($\eta$) for RE, which is scarce in the literature.

A further contribution of this work is in providing, the size effect on the mechanical response of URE. The results of the apparent Young's modulus at the wall scale were

compared to the ones at the column scale [16], which used similar soil. A close relation was observed in the evolution of the apparent Young's modulus with time between the two scales. However, differences were noted on the apparent Young's modulus when considering suction in the samples. Semi-logarithmic functions are obtained between suction and the apparent Young's modulus and the coefficient of viscosity. Similar results are obtained for soil material containing clay. All these results are promising in the understanding of the mechanical behaviour of rammed earth structures, but more tests and analyses are required to confirm the relationships between moisture content and the elasto-viscoplastic behaviour of URE.

**Author Contributions:** Conceptualization, T.C. and N.P.; Methodology, T.C. and F.A.; Validation, A.R.; Investigation, N.P.; Writing—original draft, T.C. and N.P.; Writing—review & editing, A.R. and O.P.; Supervision, O.P. All authors have read and agreed to the published version of the manuscript.

**Funding:** This research was funded by Agence Nationale de la Recherche grant number ANR JCJC VBATCH.

**Data Availability Statement:** Not applicable.

**Conflicts of Interest:** The authors declare no conflict of interest.

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
