# Peer review of "Impact of Moisture Content on the Elasto-Viscoplastic Behaviour of Rammed Earth Wall: New Findings"

_constrmater, doi:10.3390/constrmater3010001_

Round 1

Reviewer 1 Report

In the paper an experimental study to investigate the Impact of moisture content on the elasto-viscoplastic behaviour of rammed earth wall is presented.

The topic discussed in this manuscript is interesting, nevertheless it needs improvements before consideration of possible publication in the Construction materials. The paper can be accepted after MINOR REVISIONS.

        Please correct the punctuation and the English language used in the manuscript.

        In Introduction, the state of art is poor and should be expanded, in particular in line 38 and lines 43-46. As below:

  • Sabbà, M. F.; Tesoro, M.; Falcicchio, C.; Foti, D. Rammed Earth with Straw Fibers and Earth Mortar: Mix Design and Mechanical Characteristics Determination. Fibers2021, 9(5), 30.
  • Sabbà, M. F., Rizzo, F., & Foti, D. Experimental Investigation on the Compressive Strength Of Rammed Earth Panels, 2022 . Available at SSRN 4214205.
  • Maniatidis, V.; Walker, P. Structural capacity of rammed earth in compression. J. Mater. Civil. Eng. 2008, 20, 230–238.

        The formatting of the text and the images need to be corrected. Some images have not the descriptions in their bottom.

        Page 3 line 92. The model behaviour is elasto-viscoplastic, so is it possible that the two Young Modulus (E1, E2) remain constant over time? Pay attention to this point.

        Page 4 line 125. Explain if there is a specific reason why the studied wall was produced using 20 layers. Why has been this number chosen? Are there some references or regulations not cited?

        Page 4 Figure 4. The proportion of the measures indicated in the figure are incorrect, this figure does not help the reader to understand the experimental problem.

        In the paper the validation of the results of the experiments, made on two walls, is done by comparing them with the results of Chitimbo et al. (2022) who made the same experiments on URE columns. Explain the reason why it’s possible to compare the results of some experiments made on a plane element (the wall) with the results of the same experiments made on a linear element (column). The two elements have different geometry and, also, different volume so the tests made with the hydraulic hammer will certainly give different results.

Reviewer 2 Report

It's a intresting work. I think it meet the requirements of the journal. But the format of the figure should be changed.

Author Response

Comments from Reviewer 2

It's an interesting work. I think it meet the requirements of the journal. But the format of the figure should be changed.

Dear Reviewer,

Thank you very much for your time to review our manuscript and for your valuable comments which helped to considerably improve the quality of the manuscript. We have thoroughly revised the manuscript following your comment. The format of the figures is corrected, please find the modifications shown in red colored text in the revised manuscript.

Yours sincerely,

Taini Chitimbo, Noémie Prime, André Revil and Olivier Plé

Reviewer 3 Report

The manuscript presents an experimental investigation on the elasto-viscoplastic behaviour of a unstabilized rammed earth almost full-scale structural element. The results can be considered new and having in attention the green relevance of this structural material they are very relevant.

Unfortunately there are many bugs in the text, mostly caused by an insufficient domain of English. I list a few suggestions next (I will sometimes use UPPERCASE letters to highlight my suggestions):

page 1, line 10 : "This paper presents experimental test on" -> "This paper presents AN experimental test on"

p.1, l.12 : "Cyclic axial compression and stress relaxation test were carried " -> "Cyclic axial compression and stress relaxation testS were carried "

p.1, l.37 : "which is sufficient for building a few multi-storey structures" -> "which CAN BE sufficient for THE STRUCTURE OF  buildings WITH a few storeyS"

p.1, l.39 : "the mechanical behaviour of URE could vary with hydric state," -> "the mechanical behaviour of URE CAN vary with THEIR hydric state,"

p.2, l.50 (also p.7, l.209) : "viscous comportment " -> "viscous BEHAVIOUR"

p.2, l.69 : "Creep, relaxation can" -> "Creep AND relaxation can"

p.2, l.77 : "are given on Eqn. (1). Where" -> "are given BY Eqn. (1), where"

p.3, l.79 : "Although Maxwell body can describe stress relaxation" -> "Although Maxwell MODEL can describe stress relaxation"

p.3, l.81 : "more complicated models, combined with springs and dashpots, are required" -> "more complicated models, combinING SEVERAL springs and/OR dashpots, are required"

p.3, l.83 : "is the simplest to consider allowing to limit the relaxed stress to a non-zero value " - > "is the simplest ALLOWING THE STRESS TO relax to a non-zero value "

p.3, l.92 : "By substituting Eqn. (4) and (5) into Eqn. (2)" -> "By substituting Eqn. (6) into Eqn. (2)"

p.4, l.99 : "and has been used to realize" -> "and has been used to BUILD"

p.4, Figure 3 IS MISSING!

p.4 : one or two photos of the execution of the rammed earth walls in laboratory should be presented (formwork, pouring and compaction)

p.5 : one or two zoom photos of the sensors at their final position on the rammed earth wall e should be presented

p.6, l.157 (again on p.7, l.190): "The loading path is illustrated on Fig. 4:" Do the authors mean Fig.6?

p.7, Fig.8 : "its tendency shown by dotted line" -> there is no dotted line in the figure! Moreover, the caption should attribute the URE column resluts to Chitimbo et al. (2022)

p.7, l.206 : " dotted line in Fig. 9" There is no dotted line in Fig. 9. Do the authors mean dashed line?

Reviewer 4 Report

Authors present interesting experimental results on the mechanical behaviour of unstabilized rammed earth. The experimental setup and the tests performed are interesting. The influence of water content on viscous characteristics and the size effect observed are well known in literature for a variety of mechanical problems. Nevertheless, the article is interesting for the tests performed.  

Author Response

Dear Reviewer,

Thank you very much for your time to review our manuscript and for your valuable comments which helped to considerably improve the quality of the manuscript.  We have thoroughly revised the manuscript following your comments. Please find below the responses detailed; the modifications are also shown in red colored text in the revised manuscript.

Yours sincerely,

Taini Chitimbo, Noémie Prime, André Revil and Olivier Plé

Authors present interesting experimental results on the mechanical behaviour of unstabilized rammed earth. The experimental setup and the tests performed are interesting. The influence of water content on viscous characteristics and the size effect observed are well known in literature for a variety of mechanical problems. Nevertheless, the article is interesting for the tests performed. 

Answer:

-This article provides the order of magnitudes for the coefficient of viscosity(η) for RE, which is scarce to find in the literature. This sentence has been added to the conclusion of the paper.